# Non-communicable disease policy implementation in Libya: A mixed methods assessment

Luke N. Allen[1]*, Cervantée E. K. Wild[2], Giulia Loffreda[3], Mohini Kak[4], Mohamed Aghilla[5], Taher Emahbes[6], Atousa Bonyani[7], Arian Hatefi[8], Christopher Herbst[9], Haider M. El Saeh[5,6]

1 Department of Clinical Research, London School of Hygiene and Tropical Medicine, London, United Kingdom, 2 Department of Primary Care Health Sciences, University of Oxford, Oxford, United Kingdom, 3 Institute for Global Health and Development, Queen Margaret University, Edinburgh, Scotland, 4 World Bank Middle East and North Africa, Tunisia, 5 Libya National Centre for Disease Control, Tajoura, Libya, 6 University of Tripoli, Tripoli, Libya, 7 WHO Iran Country Office, Tehran, Iran, 8 World Bank Middle East and North Africa, Washington, DC, United States of America, 9 World Bank Middle East and North Africa, Riyadh, Saudi Arabia

* drlukeallen@gmail.com

**Data Availability Statement:** All data underlying our findings are available in the text and the appendix except for the interview transcripts and our notes taken during the interviews. We are not able to share these data as that would compromise

## Abstract

The Libyan Ministry of Health is keen to understand how it can introduce policies to protect its population from non-communicable diseases (NCDs). We aimed to perform an implementation research assessment of the current situation, including challenges and opportunities. We used an explanatory sequential mixed methods design. We started with a quantitative assessment of NCD policy performance based on review of the WHO NCD Progress Monitor Reports. Once we had identified Libya's NCD policy gaps we performed a systematic review to identify international lessons around barriers and successful strategies for the policies Libya has not yet implemented. Finally, we performed a series of key stakeholder interviews with senior policymakers to explore their perspectives around promising policy actions. We used a realist paradigm, methods triangulation, and a joint display to synthesise the interpretation of our findings and develop recommendations. Libya has not fully implemented any of the recommended policies for diet, physical activity, primary care guidelines & therapeutics, or data collection, targets & surveillance. It does not have robust tobacco policies in place. Evidence from the international literature and policymaker interviews emphasised the centrality of according strong political leadership, governance structures, multisectoral engagement, and adequate financing to policy development activities. Libya's complex political and security situation are major barriers for policy implementation. Whilst some policies will be very challenging to develop and deploy, there are a number of simple policy actions that could be implemented with minimum effort; from inviting WHO to conduct a second STEPS survey, to signing the international code on breast-milk substitutes. Like many other fragile and conflict-affected states, Libya has not accorded NCDs the policy attention they demand. Whilst strong high-level leadership is the ultimate key to providing adequate protections, there are a range of simple measures that can be implemented with relative ease.

the anonymity of our interviewees. The interview transcripts and our field notes both underlie our qualitative findings. The Oxford University OxTREC Ethics Committee imposed sharing restrictions on these datasets in order to protect the anonymity of the interviewees. The committee can be contacted via oxtrec@admin.ox.ac.uk.

**Funding:** This study was fully funded by the Middle East and North Africa (MENA) Transition Fund managed by the World Bank (http://seha.ly/en/) and implemented/conducted in collaboration with the Libyan Ministry of Health. The co-authors are all World Bank staff, World Bank consultants, and Libyan Ministry of Health staff – and we collectively designed the study, performed data collection and analysis, made the decision to publish, and prepared of the manuscript.

**Competing interests:** I have read the journal's policy and the authors of this manuscript have the following competing interests: Mohamed Aghilla and Haider M El Saeh hold senior positions in the Libyan National Centre for Disease Control

## Background

Non-communicable diseases (NCDs) have risen to become the leading cause of death and disability in Libya, as they are worldwide [1]. In 2019, 79% of all Libyan deaths and 78% of DALYS were caused by NCDs [2]. The 'big four' (cardiovascular diseases, cancers, chronic respiratory diseases, and diabetes mellitus) collectively account for two thirds of Libyan mortality. Ischemic heart disease and stroke have been the two top causes of deaths since 2009, and the prevalence of hypertensive heart disease and diabetes are increasing [3].

The 2009 WHO 'STEPS' survey found that 99.8% of the population had at least one of the following NCD risk factors; daily smoking; consuming fewer than five servings of fruits & vegetables per day; low levels of physical activity; overweight; or raised blood pressure [4, 5]. Dietary risks are much larger contributors to morbidity than tobacco or alcohol [2]. Obesity has more than doubled during the last three decades and over two thirds of Libyan adults are now overweight or obese [6, 7].

To address this high NCD burden, the Libyan government has endorsed the full set of WHO-recommended 'Best Buy' policies outlined in the WHO NCD Global Action Plan [8] (and summarised in Box 1), however they have not been fully implemented. These population-level interventions have a demonstrated effect size, and have been assessed for cost effectiveness, feasibility, as well as non-financial considerations in low- and middle-income countries.

## Box 1: WHO 'Best Buy' NCD Policies

1. Establish time-bound national targets based on WHO guidance.

2. Establish a functioning system for generating reliable cause-specific mortality data on a routine basis.

3. Conduct a 'STEPwise Approach' to NCD Risk Factor Surveillance survey or a comprehensive health examination survey every 5 years.

4. Develop an operational multisectoral national strategy or action plan that integrates the major NCDs and their shared risk factors.

5. Implement measures to reduce affordability by increasing excise taxes and prices on tobacco products.

6. Implement measures to eliminate exposure to second-hand tobacco smoke in all indoor workplaces, public places, and public transport.

7. Implement plain, standardized packaging or large graphic health warnings on all tobacco packages.

8. Enact and enforce comprehensive bans on tobacco advertising, promotion, and sponsorship.

9. Implemented effective mass media campaigns that educate the public about the harms of smoking, tobacco use, and second-hand smoke.

10. Enact and enforce restrictions on the physical availability of retail alcohol (via reduced hours of sale).

11. Enact and enforce bans or comprehensive restrictions on exposure to alcohol advertising (across multiple types of media).

12. Introduce excise taxes on alcoholic beverages.

13. Introduce national policies to reduce population salt and sodium consumption.

14. Introduce national policies that limit saturated fatty acids and virtually eliminate industrially produced trans-fatty acids in the food supply.

15. Implement the WHO set of recommendations on marketing of foods and non-alcoholic beverages to children.

16. Introduce legislation and regulations fully implementing the International Code of Marketing of Breast-milk Substitutes.

17. Conduct at least one recent national public awareness program and motivational communication for physical activity, including mass media campaigns for physical activity behavioral change.

18. Introduce evidence-based national guidelines, protocols, and standards for management of major NCDs through a primary care approach, that the government or competent authorities has recognized and approved.

19. Provide drug therapy, including glycemic control, and counselling for eligible persons at high risk to prevent heart attacks and strokes, with emphasis on primary care.

Source: WHO 2020 NCD Progress Monitor Report

Libya is emerging from a decade of conflict that followed the Arab Spring protests of 2011, which has left the health system fragile and deeply impacted in terms of access, service delivery and quality. For several years the country has been divided and governed by the Tripoli-based Government of National Accord (GNA) in the West and the Libyan National Army (LNA) in the East. In March 2021, through a UN facilitated peace process, a unified Government of National Unity (GNU) was established and will lead Libya until elections scheduled for 2022. The possibility of peace and stability provides tremendous opportunity for the new Government to work systematically towards strengthening and reforming health programs and policies to better align them with the current needs of the population.

To better understand the barriers, opportunities, and most appropriate approaches for implementing effective NCD policies in Libya, the Ministry of Health initiated an assessment in partnership with the World Bank, reported here. The threefold aims of this project were to understand:

1. The extent to which each of the Best Buy policies has been implemented in Libya;

2. Implementation lessons from other countries that could be used to address Libya's policy gaps;

3. Current barriers and opportunities for policy implementation as perceived by policymakers in key positions of power.

This research was funded by the Middle East and North Africa (MENA) Transition Fund managed by the World Bank and implemented/conducted in collaboration with the Libyan Ministry of Health (MoH). Methods specialists from London School of Hygiene and Tropical Medicine, the University of Oxford, and Queen Margaret University's Institute for Global Health and Development were employed to lead the project as World Bank consultants.

Professors from the University of Tripoli with key high-level policymaking experience completed the research team.

## Materials and methods

### Ethics statement

This project was initiated by the World Bank in collaboration with the Libyan Ministry of Health. The project had a very tight deadline (three months). As soon as the research team had been put together and the study protocol had been developed, our research lead (LA), World Bank team coordinator (MK) and Libyan university-based co-investigators (MES, TE) sought to obtain ethical approval from a Libyan ethics board. Very few–if any–research ethics committees are currently functioning in Libya, and none were able to review our protocol. We took advice from the Oxford Tropical Research Ethics Committee. After submitting correspondence from the World Bank (the study sponsor) confirming that it was not possible to obtain ethical approval from a Libyan board, they agreed to review the study, noting that the small number of interviews and our ethical safeguards rendered it "very low-risk". The Oxford Tropical Research Ethics Committee approved the study in July 2021 (OxTREC 541–21).

### Approach

We aimed to identify Libya's NCD Best Buy policy gaps, assess how other countries had approached similar implementation challenges, and then identify the unique policy challenges and opportunities facing Libyan policymakers. As such, we used an explanatory sequential mixed methods approach [9, 10].

### Quantitative policy review

First, we performed a quantitative assessment of Libya's NCD policy scores as reported in the 2015, 2017, and 2020 WHO NCD Progress Monitor Reports [11–13]. These reports present country-level data, scoring the level of implementation of each of 19 NCD policies ('full' implementation = 1 point; 'partial' = 0.5 points, and 'not implemented' = 0). Analysing Libya's NCD policy scores over the three years allowed us to identify four distinct groupings:

1. Policies which had been fully implemented 2015 and remained fully implemented in 2017 and 2020.

2. Policies that have been implemented since 2015.

3. Policies that have never been implemented.

4. Policies that have fallen from 'full' to 'partial/not-implemented' since 2015.

We used simple descriptive statistics to summarise the findings and compare Libya's performance to global means from the remaining 193 WHO Member States.

### Systematic review

Once we had identified Libya's NCD policy gaps we conducted a systematic review to synthesise international evidence of the mechanisms used to successfully implement these policies in other settings.

Our review was registered (PROSPERO: 42020153895) and conducted according to PRISMA and Synthesis Without Meta-analysis (SWIM) guidelines [14]. The search was conducted on 5th July 2021 using a combination of terms for NCDs and NCD Best Buy policies, on Web of Science, MEDLINE (through PubMed) Scopus, Google scholar (first 30 pages), and

the World Bank and WHO IRIS databases. Full search terms are presented in S1 Text. Results were restricted to >2011, the year when the Best Buys policies were introduced [15].

Studies that analysed adoption and implementation issues from a political economy perspective and provided empirical evidence were eligible. We excluded editorials, reviews, and conference abstracts, but performed bibliographic searches to uncover further relevant papers. Full inclusion and exclusion criteria are presented in S1 Text. One reviewer performed title and abstract screening with 20% of records independently screened by a second reviewer, using an excel spreadsheet. Dual independent review was used for full text screening, extraction, and risk of bias assessment. The reviewers resolved disagreements by consensus, with recourse to a third reviewer if necessary.

We developed an excel-based data extraction form, based on Cochrane guidance [16], which was piloted by the review team. We extracted bibliometric information, political economy factors (actors, institutions, interests, ideas and network, context, governance, power, policy dynamics, and implementation aspects), general challenges and facilitators, and specific barriers and facilitators to policy implementation relating to adoption, implementation, and adaptation to the local context [17, 18]. Our data extraction form is reproduced in S1 Text. Two independent reviewers used CERQual [19] to assess the risk of bias for each included study.

We developed a bespoke conceptual analytic framework to synthesise and analyse our findings: we set aside a series of virtual meetings between the team members and iteratively developed the key domains that emerged from the review findings. We continually refined the model until we felt that it accurately captured interrelations between the different types of outcomes. We used the model to structure our findings, and they also formed the subheadings for the interview guide used with key stakeholders.

In assessing the approaches used by other countries to address NCD policy gaps we paid particular attention to fragile and conflict-affected States [20] that may share contextual similarities with Libya. The findings from the review informed a series of targeted interviews with key NCD policymakers. We aimed to explore whether approaches used in other settings could be used domestically, and to identify major contemporary barriers and opportunities for policy implementation in Libya.

## Qualitative interviews

We used purposive sampling to recruit key policymakers and NCD policy stakeholders that had been previously identified in a separate mapping exercise conducted by World Bank and MoH staff [21]. All potential interviewees were sent information about the study via email. Participants were senior members of government and international NGOs, including representatives from the WHO, the International Rescue Committee, the National Center for Disease Control (NCDC), and the Libyan Food and Drug Control Center (FDCC). All interviewees spoke fluent English and did not require translators. Informed verbal consent was obtained from all participants and recorded using Microsoft Teams.

Five interviews were conducted by CW, MK and LA in July–September 2021 via Microsoft Teams. Open-ended and clarifying questions were used, based on a semi-structured interview guide. All interviews were audio recorded and lasted approximately 50–60 minutes. Notes were taken during the interviews and when reviewing the audio recordings afterwards.

Based on the framework constructed from the systematic review, we conducted a deductive framework analysis [22] to identify policy landscape factors and implementation challenges and opportunities under each domain, using nVivo v1.2 (QSR International Pty Ltd, Melbourne, 2020).

### Reflexivity

Our research team is multinational, with a combination of senior Libyan academic policy-makers living and working in Libya and research specialists with experience of conducting mixed-methods assessments in numerous countries but no previous links to Libya. For further detail see the COREQ checklist (S1 Checklist).

### Integration of mixed methods data

We integrated our findings from each of the three research elements using a mixed-methods joint display [23], following Guetterman and colleagues' best practice recommendations [24]. We used methods triangulation [25, 26] to explore areas of similarity and dissonance between datasets in the processes of interpretation and developing final recommendations, undergirded by a pragmatist philosophical paradigm [27, 28].

## Findings

### Quantitative policy review

By 2020, Libya had fully implemented five of the 19 WHO NCD Best Buy policies (26%): smoke-free places, tobacco advertising restrictions, alcohol sales restrictions, alcohol advertising restrictions, and alcohol taxation. Regular risk factor surveys were partially implemented in 2015 and 2017 but dropped in 2020. Clinical guidelines were partially introduced in 2020. Table 1 summarizes Libya's performance over time:

**Table 1. Implementation of the 19 WHO NCD 'Best Buy' policies over time.**

| Best Buy Policies, sorted by cluster | 2015 | 2017 | 2020 |
|---|---|---|---|
| **Targets, data collection, and plans** | | | |
| National NCD targets | 0 | 0 | 0 |
| Routine mortality data collection | 0 | 0 | 0 |
| Regular risk factor surveys | 0.5 | 0.5 | 0 |
| Multisectoral NCD plan | 0 | 0 | 0 |
| **Tobacco** | | | |
| Tobacco tax | 0 | 0 | 0 |
| Smoke-free places | 1 | 1 | 1 |
| Tobacco graphic warnings | 0 | 0 | 0 |
| Tobacco advertising restrictions | 1 | 1 | 1 |
| Tobacco mass media campaigns | N/A | 0 | 0 |
| **Alcohol** | | | |
| Alcohol sale restrictions | 1 | 1 | 1 |
| Alcohol advertising restrictions | 1 | 1 | 1 |
| Alcohol tax | 1 | 1 | 1 |
| **Diet** | | | |
| Salt reduction policies | 0 | 0 | 0 |
| Fat reduction policies | 0 | 0 | 0 |
| Child food marketing policies | 0 | 0 | 0 |
| Breast-milk substitute marketing | 0 | 0 | 0 |
| **Physical activity** | | | |
| Physical activity mass media campaigns | 0 | 0 | 0 |
| **Primary care guidelines and therapeutics** | | | |
| Clinical guidelines | 0 | 0 | 0.5 |

(*Continued*)

**Table 1.** (Continued)

| Best Buy Policies, sorted by cluster | 2015 | 2017 | 2020 |
|---|---|---|---|
| Cardiovascular therapies | 0 | 0 | 0 |
| **Total** | **5.5** | **5.5** | **5.5** |

Note: NCD, noncommunicable disease, N/A, Not Applicable; the tobacco mass media policy was only introduced in 2017

1 = fully implemented, 0.5 = partially implemented, 0 = not implemented. Scores are taken from the 2015, 2017, and 2020 WHO NCD Progress Monitor Reports

In terms of international context, Libya's overall implementation score of 5.5 places it in the lower third globally. The mean implementation score is 8.9/19 for the WHO eastern Mediterranean region, 7.6/19 for lower-middle-income countries, and 9.5/19 for upper-middle-income countries. Libya bucks the global trends for alcohol policies, with all Best Buys in place. This is a real area of strength, shared with other Middle Eastern countries with large Muslim populations.

Libya has not implemented targets or plans. These policies are among the most widely implemented policies worldwide—possibly because they require relatively low expenditures and because WHO has produced guidance, template targets, and technical support on these areas [29].

Box 2 highlights the 12 policies that have never been implemented and the single lapsed policy; together representing the country's NCD policy gaps.

## Box 2: Libya's NCD policy gaps

**Regressed since 2015**

- Regular risk factor surveys

**Never implemented**

*Targets, data collection, and plans*

- NCD targets

- Routine mortality data collection

- Multisectoral action plan

*Tobacco*

- Taxation

- Packaging graphic warnings

- Mass media campaigns

*Diet*

- Salt reduction policies

- Fat reduction policies

- Child food marketing policies

- Breast-milk substitute marketing policies

*Physical activity*

- Mass media campaigns

*Primary care guidelines and therapeutics*

- Cardiovascular risk stratification and provision of essential therapies in primary care

## Systematic review findings

Our search identified 9,659 records, of which 186 were included (Fig 1). Most studies were conducted in Europe/Americas (22%, n = 41), 6% were conducted in the Eastern Mediterranean region (n = 12), 21% in Sub-Saharan Africa (n = 40), 26% in Asia/Pacific (n = 50), and the remaining had a global or transcontinental focus. Two thirds of studies used qualitative methods (n = 120, 64%) and a quarter focused on fragile and conflict-affected settings (henceforth 'FCAS') (23%, n = 43). Almost a third of the studies focused on diet-related policies (n = 58, 31%). A full list of references included in systematic review in provided in S1 Text.

## Conceptual analytic framework

The framework that we iteratively developed to guide the organisation and analysis of our data is presented in Fig 2. Its design reflects the fact that there are a series of shared issues that relate to all Best Buy policies; then there are subsets that are unique to four different clusters: targets, data collection, and plans; primary care guidelines and therapeutics; physical activity; and a commercial determinants [30, 31] sub-cluster that included shared lessons for diet, alcohol, and tobacco. Each of these also had their own unique implementation lessons, and we also found a final subset of shared lessons that applied to diet and alcohol.

## Targets, data collection, and plans

The literature suggests that functioning NCD surveillance systems are critical for target setting, monitoring, planning, and to raise awareness and reinforce political commitments; however, these systems are often inadequate in FCAS, undermining policy planning [32–38]. Strong governance systems that facilitate multisectoral collaboration, partnership building, community mobilization, social participation, and advocacy are critical for the development of national plans, as is strong leadership to coordinate national and regional action across departments and sectors [39–45]. Under-prioritization of NCDs has resulted in insufficient resource allocation and the inability to finance national activities and plans [39, 43, 46–48]. Two well-conducted qualitative studies from Kenya, and Ghana and two mixed methods studies from Uganda and Samoa noted difficulties in shifting MoH financial resources away from historically well-funded areas (e.g. HIV/AIDS) to NCDs, making it difficult to fund surveillance activities and the development of targets and plans [33, 34, 49, 50].

**Commercial determinants of health.** There were several recurring cross-cutting issues for policies that target commercial determinants. These include legal and illegal lobbying efforts, national and transnational legal challenges, and the industry-sponsored dissemination messaging to undermine public support for new NCD policies [51–59]. Suggested actions include strategic framing of policy issues [53, 60], strengthening legal capacity to identifying

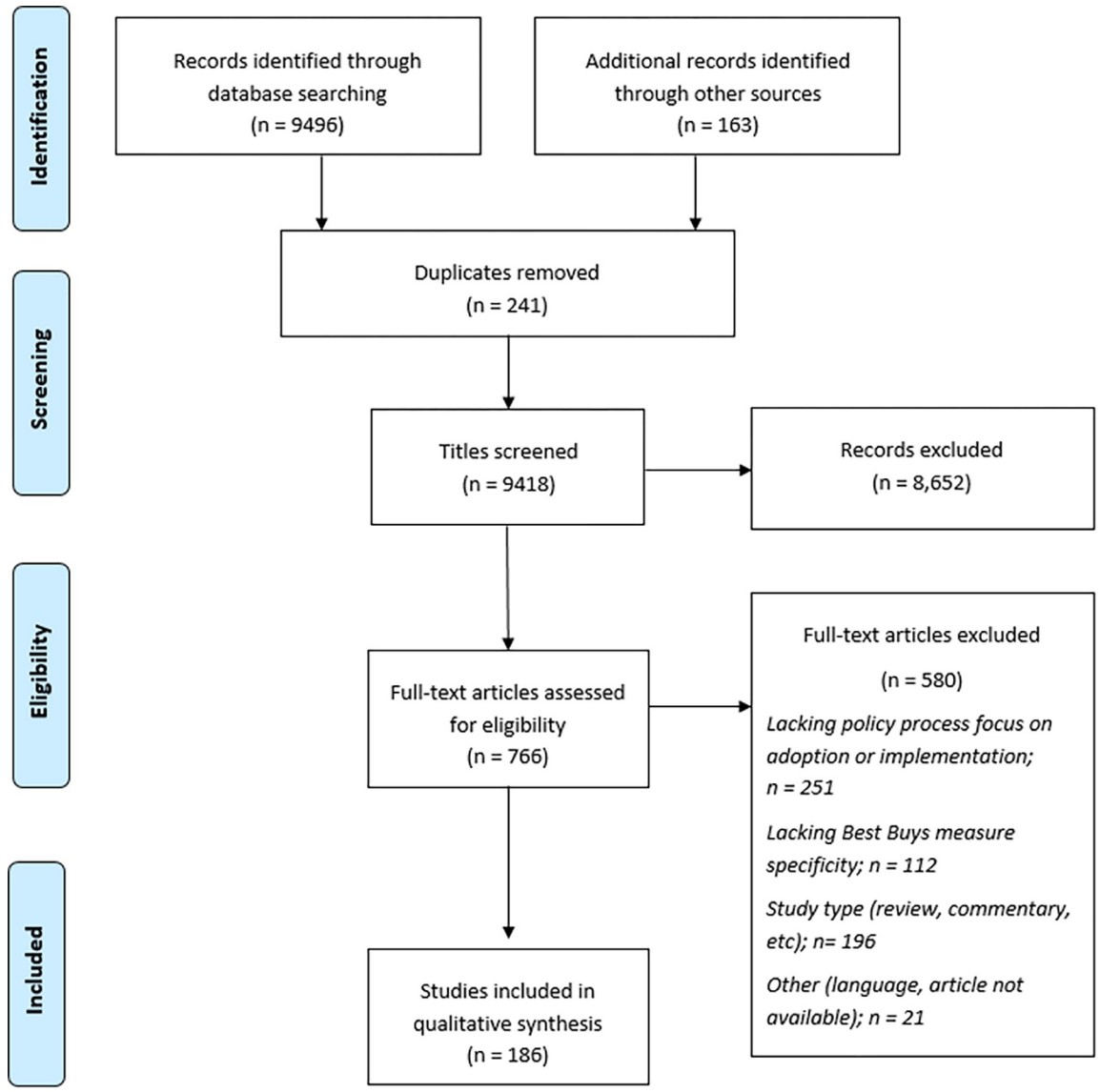

**Fig 1. PRISMA flow diagram.**

ways to minimize potential 'practical' trade concerns [54, 55], and developing clear strategies and codes of conduct to manage engagement with the private sector [61, 62].

*Tobacco*. The introduction of tobacco legislation has been heavily shaped by political, historical, social, and economic contexts [63–67]. The tobacco industry often raises considerable barriers to legislative restrictions, for instance by arguing that restrictions will increase illicit trade, create problems for retailers, harm the economy (especially in tobacco-producing countries), and violate domestic laws and international treaties on intellectual property and investments [67–71]. The Framework Convention for Tobacco Control (FCTC)—to which Libya is a signatory—has been a key tool to catalyse the process of policy formulation, adoption, and implementation [64, 72–74]. Several studies found that governance models that ensure transparent management of conflicts of interest (in line with FCTC) [67, 75, 76] and adequate funding for implementation [68, 77, 78] have been associated with implementation of tobacco

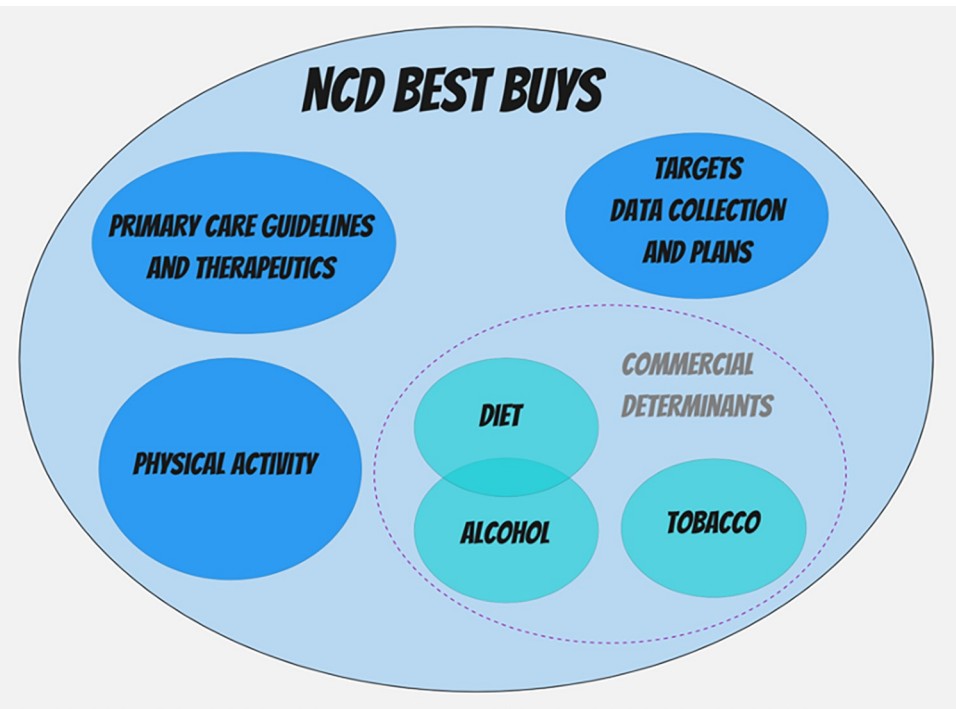

**Fig 2. Conceptual analytic framework developed to guide analysis and synthesis.**

policies. Caribbean countries that proposed harmonization of excise taxes across the region (i.e., all countries levy a minimum specific excise tax at the same rate) provided a good example of tools that enable political processes to facilitate implementation of tax policy [79].

*Diet.* In many LMICs and fragile states experiencing epidemiological and nutritional transitions, policy planning is made challenging by the fact that overweight and malnutrition can coexist [80–84]. Another commonly reported challenge is the complexity of defining clear dietary and macronutrient targets and indicators, especially given uncertainty regarding ingredients and composition of food [54, 85–90]. A perceived lack of local evidence to support links between salt or fat and NCDs are described as detrimental to the policy course [54, 85–90]. Trade challenges at the World Trade Organization or from regional agreements can exert a regulatory chill that discourages countries from implementing NCD policies [51, 91–95]. Industry also commonly seeks to forestall legislation by pushing for self-regulation [53, 61, 81, 87, 96–103]. Where adopted, effective self-regulation requires strong government leadership, multisectoral stakeholder engagement, and independent monitoring and evaluation [81, 85, 90, 96–100, 103–110].

*Alcohol.* Reported barriers to implementation include incoherent policy messaging and failure to frame the problem in a way that engenders aligned political and social understanding of the problem. In this regard, the media plays an important role in shaping public attitudes [52, 92, 93]. Two high quality country case studies reported that delayed action can stem from alcohol not being perceived as a national priority [111, 112]. Countries ran into difficulties when specific directions were not provided about how all information should be presented on labels (size, font, position, wording etc) or where WHO recommendations were not followed [113]. We note that Libya has implemented all alcohol Best Buy policies.

*Diet and alcohol.* In contrast to tobacco, we found that diet and alcohol policies were similarly undermined by a lack of global accountability instruments [51, 91–95], a lack of globally

agreed targets and evidence, and underdeveloped mechanisms for safe industry engagement [39, 41–43, 114].

*Physical Activity (PA).* Numerous social, cultural, and environmental barriers prevent engagement with PA, including fear of violence and crime in outdoor areas, air pollution, and cultural restrictions–all of which tend to disproportionately affect women and girls [115–118]. A WHO report identified that conflict, political instability, and epidemics can make it difficult to generate sufficient political attention for PA policies in many African countries [117]. Mass media campaigns are reported to be an underused tool, based on a high-quality case study from Nigeria [115]. Multisectoral collaboration was central to progressing PA policies in many settings, with successes seen when governments, NGOs, academia, transport, urban planners, and other stakeholders involved in PA promotion were brought together around shared policy goals [116, 118–120].

## Primary care guidelines and therapeutics

Fragmented health systems with a mix of private and public health provisions tend to generate a complex environment for development of national plans, particularly in fragile settings [121]. Additionally, many LMICs report competing needs from multiple disease burdens [122, 123]. Human resource capacity influences implementation: countries must train primary health care professionals and provide them with supportive materials and tools throughout the implementation period [124]. Availability of medicines and diagnostics is another major concern; this is due to limitations in procurement systems and budget allocations for medicines and blood pressure devices in some programs [50, 125]. Some population groups prefer informal providers, which adds another layer of complexity and fragmentation in health-seeking pathways [126]. Evidence on the use of traditional medicine and locally driven policies that are evidence based, cost effective, and culturally sensitive should be developed [121, 127]. Two qualitative studies conducted in Africa highlighted that the use of technology, community health workers, and task-sharing provide important frameworks to improve CVD care [128, 129].

## Interview findings

**Cross-cutting issues.** Interviewees told us that implementation of the missing Best Buy policies will only be possible with backing of political leadership at the highest level. Until new elections are held, senior policymakers are reluctant to pass major health reforms as they feel they do not have a democratic mandate. All interviewees agreed that NCD prevention and control is not a priority on the national political agenda. Persistent conflict, political instability, and the recent COVID-19 pandemic have crowded out political attention. Given the fragile security and political context, the government has limited ability to conduct any major policy reforms. Virtually all health-related policy action has focused on providing day-to-day clinical services. Coordinated responses to NCD prevention and control across the country have been challenging due to the dual governments and fragmented civil and clinical services. Overall, the security situation prevents formal engagement with national policy-making processes, impedes coordination with other sectors, and forces attention toward downstream biomedical NCD actions.

> *"Libya is in a critical situation as you know, there is political instability, you have financial problems, there is insecurity, there is incoordination between the different sectors, so all of that affects our work in NCDs and even in other areas."*

> *–Senior Government official*

**Targets, data collection, and plans.** Financing for NCD prevention and control is a challenge and the national budget does not include a specific NCD line-item, undermining both NCD surveillance and plan development. Potential funds have been diverted to the COVID-19 response and securing vaccines. Although there may be technical capacity at the ministry level to design and implement policies, there is lack of buy-in from senior decision makers. Virtually no action can be taken without approval from the very highest levels of government. One interviewee noted that even if policy makers were to pass NCD legislation, lack of enforcement capability is likely to neuter implementation.

*"There is no dedicated NCD budget. It's first come, first served."*

*–International Organization NCD lead*

Few plans relating to the Best Buys have been developed; most are in drafting stage and progress has been impeded by a lack of multi-sectoral collaboration. Much of this appears to be related to the lack of NCD surveillance.

*"I think we have not done a lot to convince the other ministries to participate with our efforts to reduce the impact of NCDs [. . .] it's very important to make people think about the size of the problem and how can be solved."*

*–Senior Government official*

In addition, ongoing conflict prevents consistent NCD surveillance. Undertaking a second STEPS survey is considered critical to persuading decision makers of the magnitude of the impact of NCDs in Libya.

*"Libya may be one of the few countries that have [not undertaken] the salt intake study. We don't know how much Libyan population are consuming salts every day. We don't have the study running yet, and this is one of the top priority research that we are thinking to conduct."*

*–Senior Government official*

**Commercial determinants of health.** Interviewees disagreed over how much corporate influence was limiting progress. Almost all processed foods, tobacco, and alcohol is imported (illegally in the case of alcohol and most of the country's tobacco) rather than produced domestically. Opportunities for regulating packaging, sales, and reformulation are perceived as minimal. One interviewee felt that health standards were not being applied consistently to imported goods. Overall, there has been little opportunity for transnational corporate interests to undermine policy because of the lack of population-level policies being proposed.

*"Big companies that are importing food. They are trying to bring something cheap to make more money and the main issue will be with the government monitoring and surveillance systems which are not active to monitor those qualities [. . .] it's mainly, I mean, laziness of those [. . .] people who are responsible for that. I don't think that it's. . . It might be corruption. I don't know. I don't see that, but it might be, I don't know."*

*–International NGO official*

**Tobacco.**   While the WHO and MoH have issued a number of decrees regarding tobacco control, none have been implemented. The relevance of tobacco control to Libyan context is somewhat disputed, with some interviewees asserting that importation and domestic supply is illegal when this is not the case. This can lead to the perception is that there is no need for tobacco-related action, despite evidence from the 2009 STEPS survey that one-quarter of the adult population smoke, and an increasing number of young people are consuming tobacco. Much of the tobacco is sold in Libya is illegal, and sold by small street vendors in the informal economy. This makes it hard to regulate.

*"The product of the tobacco industry here in Libya [is] maybe covering less than 5–10% of the daily Libyan consumption of tobacco [. . .] the main problem with tobacco is that illegal tobacco selling is rising and tobacco is available here in Libya [at a] very cheap price. . . I don't know how this tobacco enters into the country. I don't have any idea but the price of some brands of tobacco is, I think it's maybe the cheapest in the world. I mean, a packet of tobacco maybe cost around 15–20 cents. So that's very, very cheap. And as I mentioned, it's nothing to do with the with the tobacco industry, it is to do with the borders and how to control illegal tobacco trading."*

*–Senior Government official*

Responsibility for tobacco control lies at the parliamentary level, where there are multiple competing priorities. Enforcement is the responsibility of the Ministry for the Interior and the police. One interview felt that tobacco control is unlikely to be strongly enforced by police officers who smoke themselves.

*"It's of low priority, especially–I mean, who will take care of that? It is Ministry of Interior and the police people. Most of the police people, they are smoking. So how can you ask somebody to stop people from smoking while he's smoking himself? And within that facility itself, they are smoking."*

*–International NGO official*

*"They need to make some regulation, make tobacco expensive, enforce [no] smoking in public areas, there are no punishments. There are stickers on the wall, "please don't smoke" but there is no punishment for doing so. So we need strict regulation to control the tobacco in Libya."*

*–Senior Government official*

The overarching sense is that the illegal tobacco market is too large and complex to address effectively with the resources currently at the government's disposal. Furthermore, the WHO Best Buys and other policy tools are not fit for purpose as they generally do not tackle black market supply.

**Diet.**   Libya relies on food imports, with little domestic production. The FDCC applies standards that are developed by the National Center for Standardization and Metrology, however one senior policymaker stated there are no food labelling guidelines, targets or maximum thresholds for salt, fat, or sugar content in imported or domestically produced foods and beverages, which indicates that awareness of these guidelines is low. The FDCC has little capacity to develop or implement regulation of labelling guidelines or maximum thresholds for salt, fat or sugar content of foods.

There is also a paucity of country-level evidence on national diet risk factors such as salt, trans fats, and highly processed foods, which impedes action–regional or international-level evidence is perceived to be insufficient in persuading policy makers.

"*Actually, if we are thinking to take an action as you mentioned, this is not a health sector responsibility alone, it's to do with the other sectors as well. The first thing that we need to, considering this matter, is how to convince the Prime Minister to take actions [. . .] for example, to forbid any importing of trans fat products, or also to make restrictions for salt-containing foods. This is a huge responsibility shared across ministries, for example, Food and Agriculture, Prime Minister, Ministry of Health and customs [. . .] So the challenge is, how can we convince the other ministries to take such decrees to prevent to prevent importing such products*?"

*–Senior Government official*

The government has previously considered implementing the International Breastmilk code, but progress has stalled. It appears that responsibility for child health promotion is unofficially devolved to the Ministry of Education. Each time children were mentioned during the interviews, participants spoke about working in schools alongside the Ministry of Education, using education budgetary resources.

"*They [the health promotion administration] are working with the close contact with the Ministry of Education there, working at schools too and they have a clear action plan about selling unhealthy products to the school children [. . .] and there are some regulations about that. They have. . . and I can't say a proper strategy, but they have a mini strategy and action plan with the close collaboration with the Ministry of Education.*"

*–Senior Government official*

There was little concern about whether delegating all child-related issues to the education sector might weaken action to restrict the marketing of junk food to children.

**Alcohol.** Libya has fully implemented all Best Buy alcohol policies. Whilst our systematic review suggested that diet and alcohol share similar challenges in many settings, this is not the case in Libya–one of the few countries where alcohol is banned. Interviewees acknowledged that just because alcohol is banned does not mean that it is not consumed. However, WHO surveys conducted in 2008 and 2009 indicate that the bans are strictly enforced, domestic production and imports are negligible, and <1% of the population consumes alcohol. As such, alcohol is not perceived to be a legitimate risk factor of concern.

"*Alcohol is not allowed in Libya, it's illegal, so we don't have alcohol and industries here.*"

*–Government official*

Among interviewees, there was a lack of knowledge or awareness of steps that could be taken to address informally produced alcohol.

**Physical activity.** One interview participant identified Libya's lack of sports infrastructure as an impediment to state-organised physical activity promotion.

"*We don't have good spaces to implement multiple activities.*"

*–International organization official*

Developing a mass media physical activity promotional campaign through the NCD office with support from aligned NGOs would require securing a new budgetary line, which appears to be extremely difficult. A draft plan is in the early stages of development.

**Primary care guidelines and therapeutics.**   While PHC guidelines are being developed for diabetes, cardiovascular disease, hypertension, obesity, nutritional guidelines and mental health, there is little regulation across private providers. The fragmented primary health care system makes consistent implementation difficult across facilities. Conflict, political instability, and the overall lack of coordinated governance have hampered health service delivery and provision of recommended cardiovascular therapies.

*"There is no clear regulation for the private sector in Libya [. . .] to be honest, they are out of control of the Ministry of Health. So collaboration between the public and the private sector is very weak."*

*–Senior Government official*

**Synthesis and recommendations.**   We used a joint display (Table 2) to synthesise our findings, focusing on the policies that have not been implemented. Core cross-cutting themes included the strategic importance of strong governance—which equates to buy-in at the highest level of leadership for Libya, as well as adequate legislative and ministerial attention–as well as mechanisms and venues for cross-government multisectoral action that is insulated from industry interference, and dedicated funding for NCD policy development, implementation, and enforcement. The fragile security situation and fragmented health service delivery model represent major challenges for national surveillance efforts. This undermines the 'data collection' target, but also impacts other areas as policymakers feel that domestic data are required before dietary policies can be implemented. Health system fragmentation also makes it very difficult to provide consistent access to recommended cardiovascular therapies. A number of initiatives are in development, and there is a growing appreciation of the need for collaboration across government departments, however the policy stasis is unlikely to thaw until a leader with a clear electoral mandate accords NCDs the legislative priority they deserve.

## Discussion

In this explanatory sequential mixed-methods study we used quantitative policy review to identify Libya's NCD policy gaps, systematic review of the global literature to identify lessons from other settings, and key stakeholder interviews to explore specific challenges and opportunities for progress.

Libya has implemented a quarter of the 19 recommended Best Buy policies, including all alcohol policies and two tobacco policies. In reality, these tobacco measures are not enforced, smoking rates are high, and the vast majority of tobacco comes from the black market. Libya's policymakers face a unique constellation of challenges to and opportunities for introducing the NCD Best Buys. Negligible physical activity infrastructure, a world-leading illicit tobacco market, a fractured primary care system, and a decade of conflict are significant challenges. At the same time, Libya enjoys near total alcohol abstinence and reasonable political alignment regarding the need for effective NCD legislation. Perhaps the largest barrier is that new legislation requires leadership from the highest levels of government and a stable deliberative body with space on the legislative agenda. In the absence of these circumstances, policy makers have been forced to make do with downstream clinical interventions because they feel that any other form of implementation requires sign-off from an elected official.

**Table 2. Joint display of findings relating to Libya's policy gaps.**

| | Targets, data collection, and plans | Tobacco | Diet | Physical Activity | Primary Care guidelines and therapies |
|---|---|---|---|---|---|
| **Findings from quantitative policy review** | | | | | |
| *Policy gaps* | • Regular risk factor surveys<br>• NCD targets<br>• Routine mortality data collection<br>• Multisectoral action plan | • Taxation<br>• Packaging graphic warnings<br>• Mass media campaigns | • Salt reduction policies<br>• Fat reduction policies<br>• Child food marketing policies<br>• Breast-milk substitute marketing policies | • Mass media campaigns | • Cardiovascular therapies |
| **Findings from the systematic review** | | | | | |
| *Facilitators for policy implementation* | • Strong governance<br>• Multisectoral action<br>• Dedicated financing | • Dedicated financing<br>• FCTC ratification<br>• Clear governance of conflicts of interest | • Clear macronutrient & dietary targets<br>• Strong governance and multisectoral engagement<br>• Independent monitoring of voluntary industry reform/action | • Multisectoral collaboration and action | • Deployment of technology<br>• Effective exploitation of community health workers<br>• Task-sharing |
| *Challenges for policy implementation* | • Competing priorities<br>• Conflict undermining surveillance systems | • Industry opposition<br>• Legal challenges<br>• Public messaging/framing | • Industry opposition<br>• Industry self-regulation forestalling effective action<br>• Double burden of disease<br>• Insufficient local evidence on the burden of salt & fat | • Low priority issue in many settings<br>• Conflict, political instability, and epidemics all draw attention away | • Mixed public/private/traditional providers<br>• Weak health systems in fragile settings<br>• Competing priorities (HIV/Covid)<br>• Limited human resources<br>• Inadequate diagnostics Logistical and stocking issues |
| **Findings from the qualitative interviews** | | | | | |
| *Challenges for policy implementation* | • Few plans have been developed; most are in drafting stage<br>• Conflict and lack of funding prevents consistent NCD surveillance<br>• No action can be taken without sign-off from the prime minister (highest level of political leadership) | • Disputed significance; (some interviewees thought tobacco import & sale was illegal<br>• No official national tobacco strategy, although one is being drafted<br>• WHO and MoH have issued decrees regarding tobacco control; none have been implemented<br>• Counterfeit tobacco is cheap, plentiful, and sold in an informal, unregulated economy<br>• Enforcement of tobacco control challenging<br>• WHO Best Buys and other policy tools do not focus on the black market | • Reliance on imports perceived as an issue; 'it's out of our control'<br>• FDCC has low capacity to develop or implement dietary targets, labelling guidelines or maximum thresholds for salt, fat or sugar content of foods<br>• Little country-level evidence on national diet risk factors to guide action<br>• Responsibility for child health is effectively unofficially devolved to the Ministry of Education | • Paucity of sports infrastructure<br>• New promotional campaigns require new budgetary lines which are difficult to secure | • PHC guidelines being developed for diabetes, cardiovascular disease, hypertension, obesity, nutritional guidelines, and mental health<br>• Close working relationship between NCDC and primary health care division of the MoH<br>• Fragmented primary health care system makes consistent implementation difficult across facilities<br>• Little regulation or enforcement of private providers<br>• Conflict, political instability, and lack of coordinated governance have hampered health service delivery and provision of medicines |
| *Opportunities for policy implementation* | • Second STEPS survey is considered critical to persuading decision makers of the magnitude of the impact of NCDs | • [None identified] | • Functioning FDCC with good links to the NCDC<br>• Recognition that high-level leadership and multisectoral action is required | • Plan is in early stages of development | • Institute of Primary Health Care is taking responsibility for formalising treatment pathways |

Based on our findings, we have developed a set of policy recommendations for Libya. These fall into three groups according to feasibility. Firstly, simple, discrete actions that require very modest resource allocation: adding an NCD line-item to the national health budget to signal that NCDs will require dedicated resources; recognising and encouraging nascent multisectoral initiatives aimed at developing plans for physical activity via an official announcement, officially adopting the nine WHO-recommended NCD targets from the Global Monitoring Framework for NCDs (these can be updated/tailored at a later date); inviting WHO to unilaterally conduct a follow-up STEPS survey; and signing up to the International Code of Marketing of Breastmilk Substitutes. These six actions could be implemented relatively easily, particularly with the support of existing development partners in the country. They would protect lives, make good on pre-existing international commitments, and encourage encouraging policy work that is already underway. Libya's Best Buy implementation score would almost double, placing it on par with the mean for upper-middle income countries. The main barrier to action seems to be the dependence on very high-level political buy-in. Many of the interviewees noted that progress is predicated on approval at the highest level of political leadership, however, a large number of competing priorities exist in the country, many of which appear more urgent than NCD risk factor control. A further proposed issue is that the current political leadership may not feel able to introduce health policy reforms until the new elections deliver a democratic mandate to act.

The second set of recommendations require dedicated policy development: developing Libya-specific NCD targets and obtaining high-level political endorsement; drafting plain packaging legislation for legal tobacco products; setting up a commission to devise a strategy to tackle illegal tobacco; raising the official tax rate on tobacco to 75% of the total price; running mass media campaigns for a physical activity and tobacco; devising reduction and reformulation strategies for salt and fats; and developing a national policy to reduce child marketing of junk foods. The first four could be led by the NCDC. Indeed, work is already underway for many elements. The latter three (dietary) actions require ongoing collaboration between the FDCC and the NCDC, with greater involvement of the Ministry of Education. Libya already has the relevant institutional capacity, inter-sectoral relationships, and access to external expertise to deliver on all eight Best Buy domains covered above.

The third set of actions centre on structural reform. Harmonised and well-functioning national data collection systems, the reliable provision of therapeutics, and the development of a robust multisectoral action plan all require a stable security situation and political legitimacy. Our review and interviews highlight the importance of high-level political leadership for NCD policy implementation. It is clear that the role of the head of the Government is crucial for progressing nascent policy developments including to allocate hypothecated resources and engender multisectoral collaboration. The backdrop of conflict, weak governance, competing health issues (including COVID-19) and a fractured health system was found to be an important limitation in the global literature, and these issues were also felt to inhibit action in a range of areas by Libyan interviewees. These contextual issues were believed to draw legislative and fiscal attention away from NCDs as well as undermine national data collection, enforcement, and harmonised health service delivery of primary care guidelines and therapies. On one level this is understandable, however NCDs kill far more Libyans than conflict every year, and risk factor prevalence is alarmingly high. Sadly, there is little that health advocates can do to influence these macro determinants.

Our work falls under the banner of implementation research; concerned with understanding "what, why and how" interventions work for a given population in a particular setting [130, 131]. Implementation research has been defined as "the scientific study of the processes used to implement policies and interventions and the contextual factors that affect these

processes" [132]. There have been increasing calls to use IR approaches to understand how the WHO Best Buy policies can be taken to scale in low and middle income countries [131, 133–135].

## Strengths and limitations

A major strength of this study is the use of mixed-methods to gain a holistic understanding of the challenges Libya faces and the opportunities for policy progress. Our quantitative analysis followed the approach previously used by WHO and Allen et al. [29, 135, 136]. The findings are limited by the quality of the underlying Libyan Country Capacity surveys, and not all of the recommended Best Buy policies are relevant, for instance they do not capture measures to tackle illegal tobacco.

Our review was conducted according to Cochrane guidance and the PRISMA checklist. We had to relegate many details to S1 Text, however we used a robust and reproducible approach. We only used dual review for a fifth of titles & abstracts, however we erred on the side of over-inclusion and used dual review for every other step.

We used purposive sampling for our interviews and were able to access all of the senior policymakers that we approached. Again, we used a robust approach for conducting, analysing and interpreting the interviews. Ideally, we would have spoken to the higher-level political leadership (such as the health minister, finance minister, and prime minister) as they are so fundamental to the policy implementation process, but this was not thought to be feasible.

An important an overarching limitation of this work is the shifting political landscape. Whilst we deliberately spoke to policymakers who were in established positions that had survived multiple regime changes, it is possible that the institutional and legislative landscape will shift in a way that nullifies some of our recommendations. A further limitation is the narrow focus of the Best Buy policies. We sought to strike a balance between focusing on the cost-effective solutions espoused by WHO, and other NCD policies that may be more pertinent for Libya. In terms of our positionality, our research team is comprised of a mix of insiders and outsiders which provides a decent epistemological balance. The research and project leaders have clinical, policy, and research experience.

A final critical issue that our study raises is the challenge of obtaining local ethical review in fragile and conflict affected settings for short term projects that often have to deliver on very tight deadlines. We note that virtually all global health programmes run by WHO, the World Bank, and other international partner organisations operate without any form of independent ethical review. The only incentive seems to be the option of publishing findings in the peer-reviewed literature, however in an ideal world, rapid, routine processes should be in place for every project that potentially exposes participants to risk. We were unable to obtain local approval and so sought the advice of the Oxford Tropical Research Ethics Committee. They were able to provide a degree of external scrutiny. We would argue that there is a potential role for WHO to run a rapid ethics review service via a central secretariat and country representatives attached to every WHO county office. This would enable a greater proportion of non-research projects to get independent feedback on their proposed approach.

## Conclusion

NCDs are the leading cause of death and disability in Libya, however they have not been accorded adequate political attention, in part because years of conflict and constantly changing leadership inhibits action. If/when stable and legitimate national leadership emerges, it is imperative that the Government signal their blessing for greater action on NCDs. This would allow nascent collaborations and policy plans to come to fruition. Failing that, there are still a

large number of simple actions that could be approved and implemented with negligible effort. NCDs may not be as attention-grabbing as armed conflict or Covid-19, but NCDs levy an even greater burden than the two combined. What is more, the government has a range of effective tools just waiting to be used.

## Supporting information

**S1 Checklist. COREQ (consolidated criteria for reporting qualitative research).**
(DOCX)

**S1 Text. Appendix.**
(DOCX)

**S2 Text. PLOS inclusivity questionnaire.**
(DOCX)

## Acknowledgments

**Disclaimer:** The findings, interpretations, and conclusions expressed in this work are those of the authors and do not necessarily reflect the views of the Libyan Ministry of Health or the World Bank, their Boards of Directors, or the governments they represent.

## Author Contributions

**Conceptualization:** Mohini Kak, Mohamed Aghilla, Taher Emahbes, Arian Hatefi, Christopher Herbst, Haider M. El Saeh.

**Data curation:** Giulia Loffreda.

**Formal analysis:** Luke N. Allen, Cervantée E. K. Wild, Giulia Loffreda, Mohini Kak, Atousa Bonyani.

**Funding acquisition:** Mohini Kak, Mohamed Aghilla, Christopher Herbst, Haider M. El Saeh.

**Investigation:** Luke N. Allen, Cervantée E. K. Wild, Giulia Loffreda, Taher Emahbes.

**Methodology:** Luke N. Allen, Cervantée E. K. Wild, Giulia Loffreda, Mohini Kak.

**Project administration:** Luke N. Allen, Giulia Loffreda, Mohini Kak, Mohamed Aghilla, Taher Emahbes, Arian Hatefi, Christopher Herbst, Haider M. El Saeh.

**Resources:** Luke N. Allen, Cervantée E. K. Wild, Giulia Loffreda, Mohini Kak.

**Software:** Luke N. Allen, Giulia Loffreda.

**Supervision:** Luke N. Allen, Mohini Kak, Arian Hatefi, Christopher Herbst.

**Validation:** Luke N. Allen, Cervantée E. K. Wild, Giulia Loffreda.

**Visualization:** Luke N. Allen, Giulia Loffreda.

**Writing – original draft:** Luke N. Allen, Cervantée E. K. Wild, Giulia Loffreda.

**Writing – review & editing:** Luke N. Allen, Cervantée E. K. Wild, Giulia Loffreda, Mohini Kak, Mohamed Aghilla, Taher Emahbes, Atousa Bonyani, Arian Hatefi, Christopher Herbst, Haider M. El Saeh.

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
