## [Decision Letter · Decision Letter 0]

10 Jun 2022

PGPH-D-22-00134

Non-Communicable Disease Policy Implementation in Libya: a mixed methods assessment

Dear Dr. Allen,

Thank you for submitting your manuscript to PLOS Global Public Health. After careful consideration, we feel that it has merit but does not fully meet PLOS Global Public Health’s publication criteria as it currently stands. Therefore, we invite you to submit a revised version of the manuscript that addresses the points raised during the review process.

The manuscript has been evaluated by one reviewer, and his comments are available below.

The reviewer has raised a number of concerns. He requests improvements to the reporting of methodological aspects of the study (such as information on alcohol use), revisions to the systematic review findings and he requests to improve the policy recommendation and results section.  He also has concerns about the language used and request copy editing.

Could you please carefully revise the manuscript to address all comments raised?

We look forward to receiving your revised manuscript.

Kind regards,

Lorena Verduci

Staff Editor

Journal Requirements:

2. Please send a completed 'Competing Interests' statement, including any COIs declared by your co-authors. If you have no competing interests to declare, please state "The authors have declared that no competing interests exist". Otherwise please declare all competing interests beginning with the statement "I have read the journal's policy and the authors of this manuscript have the following competing interests:"

3. Please amend your detailed Financial Disclosure statement. This is published with the article. It must therefore be completed in full sentences and contain the exact wording you wish to be published.

a. Please clarify all sources of funding (financial or material support) for your study. List the grants (with grant number) or organizations (with url) that supported your study, including funding received from your institution. 

b. State the initials, alongside each funding source, of each author to receive each grant.

c. State what role the funders took in the study. If the funders had no role in your study, please state: “The funders had no role in study design, data collection and analysis, decision to publish, or preparation of the manuscript.”

4. Please provide separate figure files in .tif or .eps format and removed from the manuscript file.

5. We have noticed that you have uploaded Supporting Information files, but you have not included a list of legends. Please add a full list of legends for your Supporting Information files after the references list. 

Additional Editor Comments (if provided):

Reviewers' comments:

Reviewer's Responses to Questions

**Comments to the Author**

1. Does this manuscript meet PLOS Global Public Health’s publication criteria? Is the manuscript technically sound, and do the data support the conclusions? The manuscript must describe methodologically and ethically rigorous research with conclusions that are appropriately drawn based on the data presented.

Reviewer #1: Partly

2. Has the statistical analysis been performed appropriately and rigorously?

Reviewer #1: N/A

3. Have the authors made all data underlying the findings in their manuscript fully available (please refer to the Data Availability Statement at the start of the manuscript PDF file)?

Reviewer #1: Yes

4. Is the manuscript presented in an intelligible fashion and written in standard English?

Reviewer #1: Yes

5. Review Comments to the Author

Reviewer #1: Thanks for the opportunity of reviewing this interesting, and on the whole, well written manuscript.

The manuscript does not have page numbers (or line numbers) which is an inconvenience for reviewing. Please make sure that you've complied with the instructions for authors found at https://journals.plos.org/globalpublichealth/s/submission-guidelines

Please also conform to reporting guidelines for qualitative research, for example https://www.equator-network.org/reporting-guidelines/coreq/

There is a grammatical error in the sentence - "We developed a bespoke conceptual analytic framework to synthesise and analysis of our findings"

There is a "t" missing from "brought" in the sentence "Multisectoral collaboration was central to progressing PA policies in many settings, with successes seen when governments, NGOs, academia, transport, urban planners, and other stakeholders involved in PA promotion were brough together around shared policy goals."

Some content in the methods should be put in the results instead - e.g. the conceptual framework that guided ongoing analysis, this is itself part of the findings of this study.

SR-6% of studies found were from the Eastern Mediterranean, not 0.6% as stated

Just because alcohol is illegal does not mean it is not consumed. I think that this needs to be more fully discussed in the manuscript.

I find "The only material barrier is prime ministerial bandwidth and attention" a very interesting sentence. Firstly, "bandwidth" is a very informal term when used in this context - what exactly does it mean? This feels more like editorializing than scientific writing. Also - as an outsider to the context, this seems to be quite a political statement, perhaps you could comment on this?

Same comment for "Until Libya elects a prime minister who is willing to green-light nascent policy developments, allocate hypothecated resources, and engender multisectoral collaboration, our research suggests that progress is very unlikely" - this is apparently a very political comment, please provide a clearer justification for this.

It's not clear to me why your policy recommendations come before the discussion section.

6. PLOS authors have the option to publish the peer review history of their article (what does this mean?). If published, this will include your full peer review and any attached files.

**Do you want your identity to be public for this peer review?** For information about this choice, including consent withdrawal, please see our Privacy Policy.

Reviewer #1: **Yes: **M D Bould

---

## [Editor Report · Decision Letter 1]

11 Oct 2022

Non-Communicable Disease Policy Implementation in Libya: a mixed methods assessment

PGPH-D-22-00134R1

Dear Dr Allen,

We are pleased to inform you that your manuscript 'Non-Communicable Disease Policy Implementation in Libya: a mixed methods assessment' has been provisionally accepted for publication in PLOS Global Public Health.

Best regards,

Madhukar Pai, MD, PhD

Editor-In-Chief
